# Design of a Wide-Dynamic RF-DC Rectifier Circuit Based on an Unequal Wilkinson Power Divider

**Cheng Peng** [1,2] , **Zhihao Ye** [1] , **Jianhua Wu** [1,*] , **Cheng Chen** [1] and **Zerun Wang** [1]

1   College of Electrical Engineering, Naval University of Engineering, Wuhan 430010, China; raul0421@sina.com.cn (C.P.); yxyx928@126.com (Z.Y.); chencheng_wpt@whu.edu.cn (C.C.); wzrunfly@163.com (Z.W.)
2   College of Information and Communication, National University of Defense Technology, Wuhan 430010, China
*   Correspondence: jianhuafly@163.com

**Abstract:** In this paper, a dual-channel RF-DC microwave rectifier circuit is designed with a 2:1 power distribution ratio in a Wilkinson power splitter. The rectifier circuit works at 2.45 Ghz. After impedance matching and tuning, the structure is able to broaden the dynamic power range of the rectifier circuit while maintaining maximum rectifier efficiency. Compared with the HSMS2820 rectifier branch, this design enhances the power dynamic ranges of 60% efficiency and 50% efficiency by 4 dBm and 3 dBm, respectively. Compared with the HSMS2860 rectifier branch, for the efficiency of 60% and efficiency of 50%, the power dynamic range is expanded by 5 dBm and 2 dBm, respectively. This shows that the technology is helpful for improving the stability of energy conversion at the receiver end of microwave wireless energy transmission systems. Finally, the rationality of this conclusion is verified by establishing a mathematical model.

**Keywords:** unequal Wilkinson power divider; wide dynamic; RF-DC microwave rectifier circuit

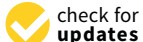



## 1. Introduction

Wires have always acted as carriers for power transmission since the invention of electric energy. However, with technical progress and the expansion of the human sphere of activities, power transmission through wires is no longer able to satisfy special power demands anymore (such as autonomous charging robots, satellites, etc.). Under these circumstances, Wireless Power Transfer (WPT) emerged at the right moment [1]. Relative to Inductive Power Transfer (IPT) [2], Resonant Power Transfer (RPT) [3] and Microwave Wireless Power Transfer (MWPT) possess the features of remote transmission distance, high power, and certain penetrability, and therefore are being used by an increasing number of people. In particular, with the development of new energy technology and people's focus on solar energy, the advantages of MWPT with respect to remote transmission distance will play an important role in power transmission for solar power stations (SPSs) [4]. Moreover, devices can be located near sources of radiation, such as base stations, to recycle electromagnetic energy. In MWPT systems, a microwave rectifier circuit at the receiver end is able to convert radio frequency (RF) energy to direct current (DC) energy, playing a crucial role, as shown in the red box in Figure 1.

In practical applications, the overall efficiency of an MWPT system's recruiter circuit at the receiver end is extremely unstable, because the RF power received by the receiving antennas will be affected by topography, landform, ground features, transmission paths, and polarization matching. When the input power is changed, great changes in input impedance will occur due to the nonlinearity of the rectifying devices, resulting in impedance mismatch and an extreme reduction in rectification efficiency. To overcome such issues, wide-dynamic RF-DC microwave rectifier circuits have been the subject of

much study by domestic and overseas scholars. The so-called wide-dynamic RF-DC mi-crowave rectifier circuit maintains RF-DC conversion efficiency at a higher level when the input power changes over a wider scope. According to the existing literature, some progress has been made in this field. For example, one scholar used MOSFET's sensitivity to voltage to design two lines of self-adaptive control rectifier circuit [5]. Switching the field-effect tube control circuit between series and parallel expands the working range of circuit power. Even if the principle of the power control technology is simple, this enhances the microwave rectifier circuit's adaptability to different levels of input power to some extent. Due to the introduction of a complicated control circuit or network like MOSFET, some input power is consumed, leading to the efficiency of the rectifier circuit being affected. For this reason, one scholar proposed a design method for wide-dynamic rectifier circuits based on passive microwave networks. A scholar from the Massachusetts Institute of Technology (MIT) put forward the real impedance compression technique [6]. First of all, the changes in the rectifier circuit's impedance with input power are adjusted to the real impedance through the differential circuit, and then the change scope is declined. This method was able to successfully achieve wide-dynamic rectification, within certain limitations. In the last two years, multi-channel output and energy recycling technologies based on mismatching properties have become a research hot spot. Based on two lines of a microwave rectifier circuit, a research team at the South China University of Technology recycled mismatching reflective energy through a directional coupler and enhanced the rectifier circuit's dynamic scope for power response [7].

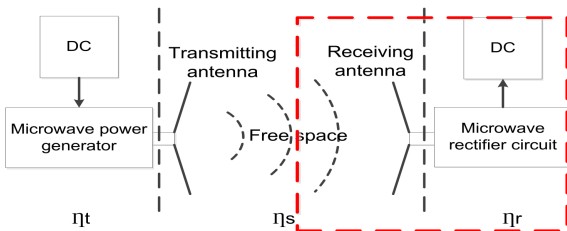

**Figure 1.** Schematic diagram of an MWPT system.

A microwave network for a Wilkinson power divider is used in this paper to expand the dynamic power range of rectifier power. RF power distribution technology is adopted to divide the input RF energy into two lines of 2:1. One line is an RF-DC circuit with good high-power response, while the other line is an RF-DC circuit with good low-power response. While a similar design was reported in [8], the implementation in this paper is different; the former was based on three band pass filters acting as a dc filter to suppress harmonics produced by nonlinear devices, while this paper adopts two levels before and after impedance matching to ensure the transmission efficiency of the whole circuit, thus maintaining a higher work frequency, which is the most important aspect. Most studies only describe the engineering implementation method, and provide a simple qualitative description. In this paper, an approximate mathematical model is established to explain the real reason this method achieves a wide dynamic. Because of the characteristics of the dual rectifier circuit, when the input power changes dynamically, the overall RF-DC conversion efficiency remains high [9–11].

## 2. Single RF-DC Circuit Design

Before designing the dual RF-DC microwave rectifier circuit, a single RF-DC mi-crowave rectifier circuit was designed using Advanced Design system (ADS) 2017 SOFT-WARE, using the structure shown in Figure 2. The single RF-DC microwave rectifier circuit is the most traditional RF-DC rectifier, wherein a receiving antenna converts microwave energy into RF energy in the microwave circuit and then converts it to DC energy by means of the rectifier network, which consists of a Schottky diode, which is then delivered to the load [12–18]. Given that a diode is a nonlinear component and produces higher harmonics,

a DC filter needs to be introduced between the rectifier network and the load in order to inhibit the harmonic influence and prevent high-frequency energy leakage [19–24]. To enhance transmission efficiency, an impedance matching network should be connected between the rectifier circuit and the antenna in order to reduce high-frequency energy reflection. Sometimes, it is also necessary to perform impedance matching separately between the DC filter and the load.

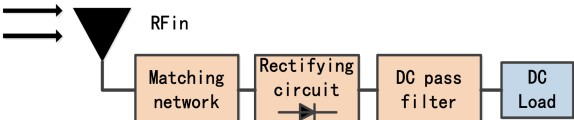

**Figure 2.** Structure diagram of the single microwave rectifier circuit.

### 2.1. Design of the DC Filter

The DC filter is an important component of microwave energy collection systems. As shown in the microwave energy collection system drawing in Figure 2, the DC filter is situated between the RF-DC microwave rectifier circuit and the end load. Since the Schottky diode is a nonlinear component, higher harmonics will be formed during the rectification process, thus affecting the rectifier waveform and causing additional energy loss. Generally speaking, such higher harmonics occur in multiples of two of the resonant frequent. For instance, when the resonant frequency is 2.45 GHz, the primary harmonic occurs at 4.9 GHz. At the same time, such higher harmonics result in additional energy loss while also impacting the load voltage. The DC filter enables DC to pass, while the higher harmonics generated by the RF-DC microwave rectifier circuit are reflected back into the diode for rectification until they arecompletely converted into a DC component, strongly enhancing rectification efficiency.

Low-pass or band elimination is available in the direct filter. Given that the primary, secondary, and tertiary harmonics are only generated at 2.45 GHz, 4.9 GHz, and 7.35 GHz, if a band elimination filter is adopted, it will further enhance the harmonic reflectivity, thus improving rectification efficiency. To this end, a three-band elimination filter was applied in this paper to filter the primary, secondary, and tertiary harmonics, as shown in Figure 3. Port 1 was connected to the output of the diode rectifier circuit, and port 2 was connected to the load; the thickness of the dielectric plate $H = 0.762$ mm, and the dielectric constant $\varepsilon_r = 2.65$. On the basis of simulation optimization, the sizes of each microstrip branch of the filter circuit were determined to be $L1 = 3.75$ mm, $L2 = 5.4$ mm, $L3 = 12$ mm, $L4 = 3.4675$ mm, $L5 = 10$ mm, $L6 = 1.7675$ mm, and $L7 = 6$ mm; among these, the three short parallel branches adopted a 70-degree fan structure to improve their impedance matching performance.

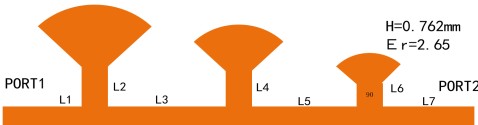

**Figure 3.** Structure diagram of the three-band filter.

It can be seen from the S11 parameters of the three-band filter presented in Figure 4 that the filter is equipped with remarkable trafficability at the DC part, while the S21 parameter clearly shows that the values at the primary harmonic 2.45 GHz, the secondary harmonic 4.9 GHz, and the tertiary harmonic 7.4 GHz reach −40 dBm, −60 dBm, and −30 dBm, respectively, thus better eliminating each harmonic's influence on the rectifier circuit and demonstrating excellent traceability and harmonic inhibiting ability. The filter re-reflects the generated harmonic back to the diode rectifier circuit, where it continues to participate in the rectification process until it completely becomes DC, greatly improving the RF-DC conversion efficiency of the circuit. At the same time, when the frequency

is 0, the insertion loss of the filter is only −1.115 dB, so the filter demonstrates good DC conduction performance.

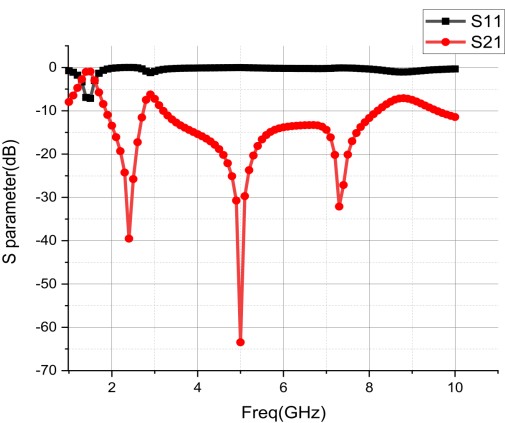

**Figure 4.** S parameter simulation of the three-band elimination filter.

## 2.2. Single Matching and Optimization

The designed filter is inserted between the Schottky diode and the load. Because impedance matching is conducted by the band elimination filter on the basis of 50 Ω input impedance, a part of stub line is added in parallel between the Schottky diode and the three-band DC filter in order to perform impedance matching at the rear end. Thus, the diode's rectifier circuit, the DC filter, and the load can be viewed as load impedance as a whole, while the stub lines in series and parallel should be added at the input port and rectifier circuit in order to perform front-end impedance matching, thereby optimizing the RF-DC conversion efficiency of the entire rectifier circuit, as shown in Figure 5:

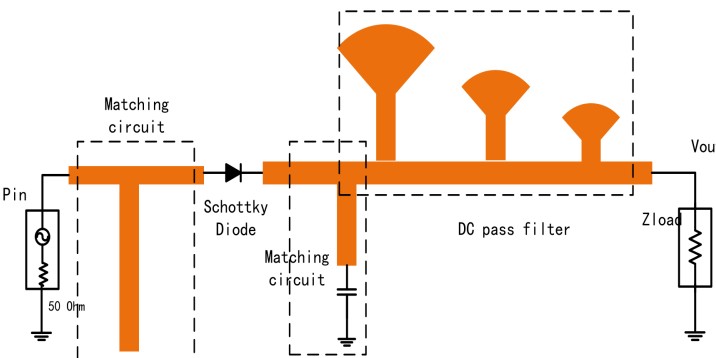

**Figure 5.** Structure diagram of the single microwave rectifier circuit.

It is worth noting that, when adjusting impedance matching, the overall input impedance at the rear-end circuit needs to be measured during the process, and the size of impedance stub should then be confirmed on the basis of the rear-end input impedance, because it is necessary to perform matching for two-line rectifier circuits in different situations. Equipped with the unequal power divider, the HSMS2820 diode, possesses a good response to high-power inputs, was applied on one circuit, while, due to the low power in the other circuit, the HSMS2860 diode, which possesses a good response to low-power inputs, was adopted therein. Table 1 shows the performance parameters of the two types of diode.

**Table 1.** Performance parameters of HSMS2820 and HSMS2860 Schottky diodes.

| Types | Ohmage | Backward Voltage | Offset Capacitance | Saturation Current | Launching System | Range |
|---|---|---|---|---|---|---|
| HSMS2820 | 6 | 15 | 0.7 | $2.2 \times 10^{-8}$ | 1.08 | >−20 dBm |
| HSMS2860 | 5 | 7 | 0.18 | $5 \times 10^{-6}$ | 1.08 | >−20 dBm |

It is worth emphasizing that due to the use of different Schottky diodes in the two-line rectifier circuit, impedance matching needs to be optimized individually for each circuit. In fact, different diodes exhibit different impedance characteristics under the same conditions. To optimize the overall effect of the circuits, it is necessary to optimize the parameter matching of the two-line diode rectifier circuits. For the purpose of verifying the rectification effect of each circuit, the input power was respectively set as 6 dBm, 7 dBm, 8 dBm, 9 dBm, 10 dBm, 11 dBm, and 12 dBm, respectively, the voltage curve of each circuits output as a function of time was obtained, as shown in Figure 6:

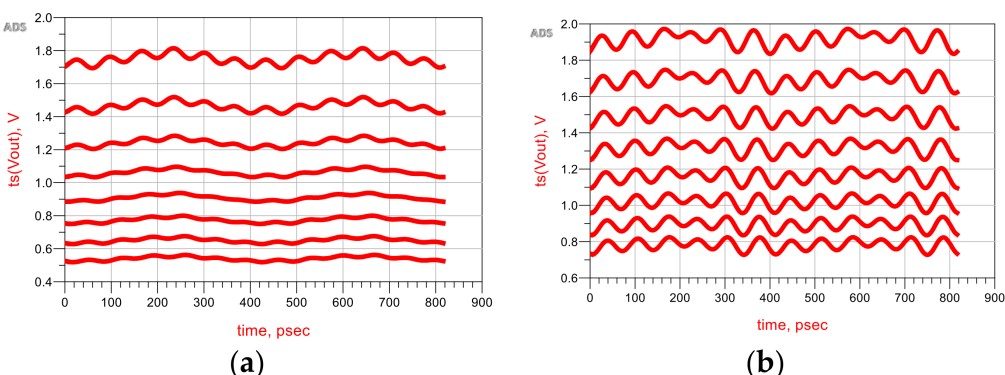

**Figure 6.** Single rectifier load voltage–time curve. (**a**) The load voltage–Time curve of the HSMS2820 Circuit; (**b**) the load voltage–time curve of the HSMS2860 circuit.

The curve in Figure 6 indicates that the output voltage in each circuit excludes negative values. After confirming input power Pin, output voltage Vout basically remains at a fixed value, indicating that this circuit has achieved RF-DC rectification. Moreover, the higher the Pin is, the larger the output voltage Vout will be. When input power is close to but does not reach reach the maximum rectification efficiency, voltage ripple disturbance increases. When close to the maximum rectification efficiency, the diode conversely breaks down the disturbance triggered by this effect. The small overall sampling values are further from the maximum rectification efficiency value of the HSMS2820. Therefore, with respect to sampling power, the output voltage of the HSMS2820 circuit is more stable than the output voltage of the HSMS2860 circuit.

Input power Pin (unit dBm) is regarded as a variable. After conversion, the actual input power $P = 10^{P_{in}/10}$ (unit mW) is obtained, and the real component of the output voltage Vout should be Vr (unit V), while Zload denotes the load impedance. The PCE of the efficiency of single RF-DC microwave rectifier circuit can be calculated using the following formula:

$$P_{CE} = \frac{1000 \cdot V_r^2 / Z_{load}}{P} \times 100\%$$

The single rectification efficiency–input power curve following optimization is illustrated in Figure 7, showing that the HSMS2820 diode performs with good efficiency for higher levels of input power. When input power, Pin, is 23 dBm, the maximum RF-DC rectification efficiency is 77.55%. When $P_{in} \in [16, 25]$, the rectification efficiency is greater than 60% for a dynamic range of 9 dBm. When input power $P_{in} \in [13, 26]$, the rectification efficiency is greater than 50% over a dynamic range of 13 dBm. It can be observed from Figure 7b that the HSMS2860 diode shows a favorable response for lower levels of input power. When input power, Pin, is 13 dBm, the maximum rectification efficiency is 77.66%. When $P_{in} \in [4, 14]$, the rectification efficiency is greater than 60% over a dynamic range of 10 dBm. When $P_{in} \in [0, 16]$, the rectification efficiency is greater than 50% over a dynamic range of 16 dBm. The circuit layout of each branch is simulated in this paper, and the EM simulation curve is consistent with the schematic diagram.

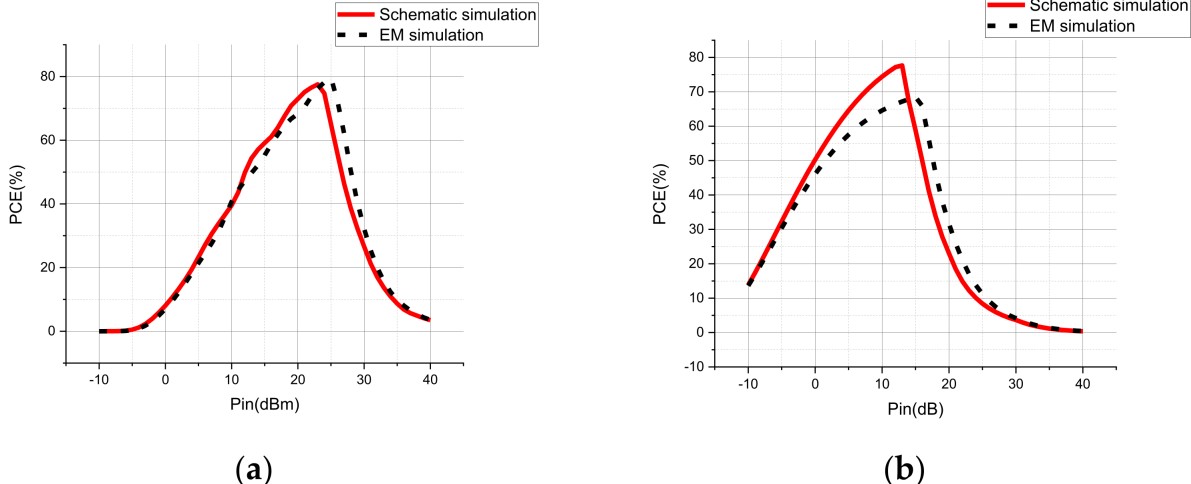

**(a)**                                                                **(b)**

**Figure 7.** Single RF-DC circuit efficiency–input power curve. (**a**) Rectification efficiency–input power curve of the HSMS2820 circuit; (**b**) rectification efficiency–input power curve of the HSMS2860 circuit.

## 3. Design of the Dual RF-DC Circuit

### 3.1. Design of the Unequal Power Divider

To distribute the RF power at a ratio of 2:1, a 2:1 power distribution network based on the Wilkinson power divider was designed using simulation software. The schematic diagram of the circuit is illustrated in Figure 8:

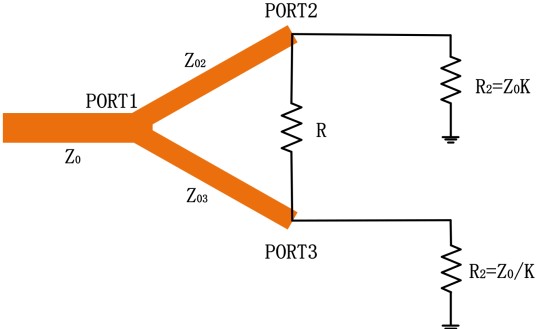

**Figure 8.** Schematic diagram of the 2:1 Wilkinson power divider.

Assuming that the power ratio between port 2 and port 3 was K2 = P3/P2, the following design formula was applied. As K = 1, this circuit is a power divider circuit, wherein it can be observed that the output line matches impedance R2 = Z0K and R3 = Z0/K, instead of being matched with impedance Z0. A matching convertor is available to convert such output impedances. When $K = 1/\sqrt{2}$, the output power at port 2 is exactly double that at port 3.

$$Z_{03} = Z_0 \sqrt{\frac{1+K^2}{K^3}}$$
$$Z_{02} = K^2 Z_{03} = Z_0 \sqrt{K(1+K^2)}$$
$$R = Z_0 \left(K + \frac{1}{K}\right)$$

In accordance with this principle, the unequal power divider was designed on the basis of simulation. After determining the impedance characteristic, a plate with a thickness of H = 0.762 mm and a dielectric constant of $\varepsilon_r = 2.65$ were selected. By means of software simulation, the S parameter curve of the unequal power divider was obtained as shown in Figure 9. The work frequency reflectance of port 1 of this power divider at 2.45 GHz was S11 = −28.747 dB. Then, the reflectance at port 2 and port 3 were determined to be S22 = −33.608 dB and S33 = −35.803 dB, respectively, implying that this power

divider was able to achieve better impedance matching at the working frequency of 2.45 GHz, demonstrating excellent transmission efficiency. The insertion loss between port 1 and port 2 was S21 = −1.742 dB, and the insertion loss between port 3 and port 2 was S31 = −4.966 dB, representing a difference of 3.224 dB. This result indicates that the output power at port 2 was twice that at port 3, representing power matching reflecting the 2:1 power divider ratio.

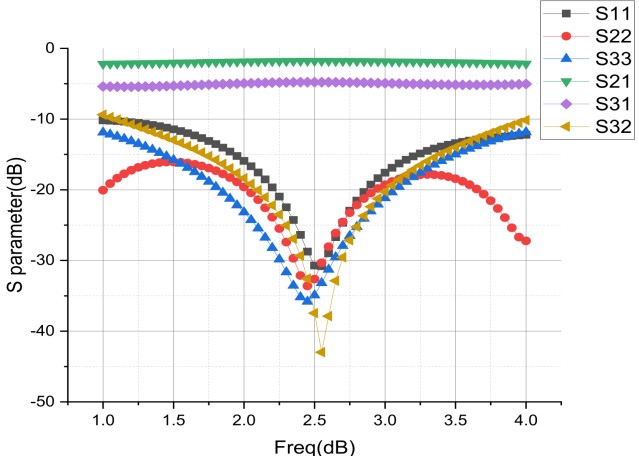

**Figure 9.** S parameter curve of the 2:1 Wilkinson power divider.

### 3.2. Overall Circuit Debugging

The single RF-DC microwave rectifier circuit is linked to HSMS2820 diode rectifier circuit I, which possesses a good response to high input power, at port 2, and HSMS2860 diode rectifier circuit II at port 3 of the power divider. Given that the power divider introduced will affect the overall matching characteristics of the circuit, each branch was adjusted to the matching state, but they cannot be adjusted too dramatically. The fine-tuning optimization method was applied for the load in order to adjust the load in the upper and lower circuits to different values [25–27]. The optimized loads of rectifier circuit I and rectifier circuit II were confirmed as Zload1 = 430 Ω and Zload2 = 260 Ω, respectively, demonstrating the excellent overall performance of the circuit. The system's resonant frequency was still 2.45 GHz, the medium thickness was H = 0.762, the dielectric constant was $\varepsilon_r = 2.65$, the microstrip line texture was copper, the thickness was T = 0.035 mm, and the loss angle tangent was set as tanD = 0.001. The schematic diagram and layout of the dual RF-DC microwave rectifier circuit based on the unequal power divider was obtained on the basis of a simulated design, as shown in Figure 10:

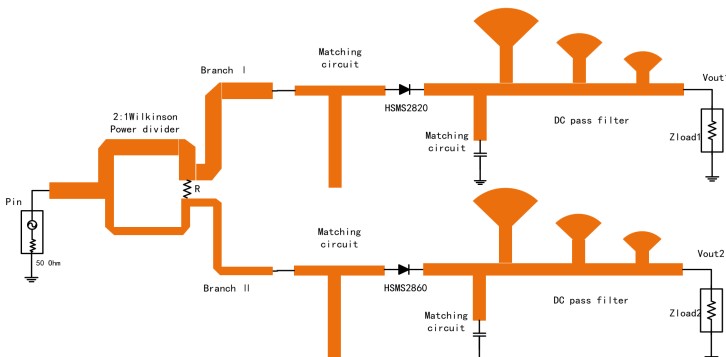

**Figure 10.** The schematic diagram of the 2:1 dual RF-DC microwave rectifier circuit.

To analyze the relationship between rectification efficiency and input power, the load and real output voltage at circuit 1 were respectively set up as Zload1 and Vout1

(unit V). Then, the load and real output voltage at circuit 2 were established as Zload2 and Vout2 (unit V), respectively. Even if output voltage passes through DC filtering, there are still fewer high-frequency components (DC voltage ripples). Since such high-frequency components cause an extremely small impact, they can be ignored. The rectification efficiency of the entire circuit is PCE, and the input power is Pin (unit dBm). Through conversion, an actual input power of $P = 10^{P_{in}/10}$ (unit: mW) is obtained. Then, the total efficiency of the dual RF-DC microwave rectifier circuit is calculated on the basis of the following formula:

$$P_{CE} = \frac{(V_{out1}^2/Z_{load1} + V_{out2}^2/Z_{load2}) \cdot 10000}{P} \times 100\%$$

The input power was set up as Pin (dBm). The dynamic range of the Pin variable parameters in the high-frequency circuit simulation software was set up as −10 dBm–40 dBm. The efficiency–input power relationship curve of the dual RF-DC rectifier circuit based on the unequal power divider is plotted in Figure 11, in accordance with the calculation formula. To prove the realizability of the circuit, the EM rectifier circuit is also drawn for the purposes of comparison.

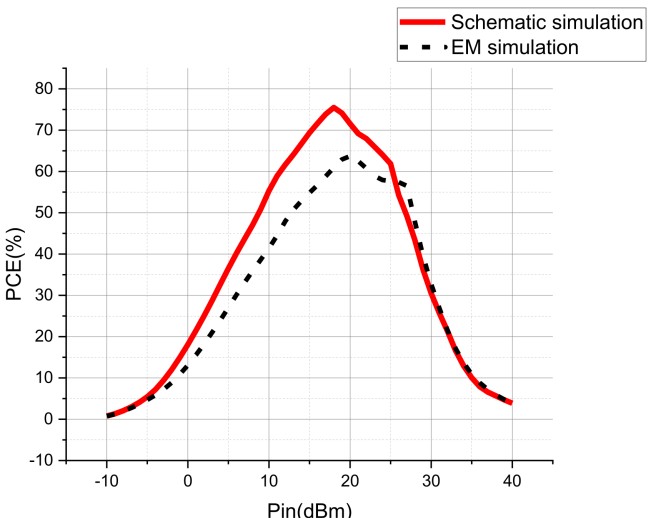

**Figure 11.** Transmission efficiency comparison of the single/dual RF-DC microwave rectifier circuit.

It can be observed from the simulation curve that the maximum rectification efficiency, which was 75.49%, for the 2:1 dual RF-DC microwave rectifier circuit occurred at Pin = 18 dBm. At Pin = 13 dBm, within the dynamic range of 12–25 dBm, the RF-DC rectification efficiency was greater than 60%. At Pin = 18 dBm within the dynamic range of 9–27 dBm, RF-DC rectification efficiency was greater than 50%. The EM simulation curve is basically consistent with the schematic diagram simulation, proving the feasibility of the design, while the errors related to solder joints and holes are still unavoidable. The maximum rectification efficiency obtained by EM simulation was only 63.7%, which is slightly lower than the maximum rectification efficiency obtained with the schematic diagram simulation.

## 4. Comparison Analysis

To better clarify the influence of the dual RF-DC microwave rectifier circuit based on an unequal Wilkinson power divider on the dynamic scope of circuit power, the 2:1 dual rectifier circuit was compared with the dynamic power range of the HSMS2820 circuit and the HSMS2860 circuit. The curves of the rectification efficiency of three types of rectifier circuit PCE changing with input power, Pin, are plotted in Figure 12. It can be observed from the maximum rectification efficiency that the efficiency of 2:1 dual RF-DC rectification

is slightly lower than that of single ones, because the complicated circuit structure results in additional energy loss (such as thermal loss of circuits). However, within the dynamic power range, only 13 dBm of the dynamic power range achieves more than 60% of 2:1 dual RF-DC rectification efficiency. Compared to the HSMS2820 circuit, this is an increase of 4 dBm, while relative to HSMS2860, it is an increase of 3 dBm. A value of 18 dBm was obtained for the dynamic power range above 50%, which is an increase by 5 dBm with respect to the HSMS2820 circuit, and an increase by 2 dBm relative to HSMS2860. The simulation result demonstrates that relative to the traditional single RF-DC microwave rectifier circuit, this design is able to remarkably widen the dynamic RF-DC power range.

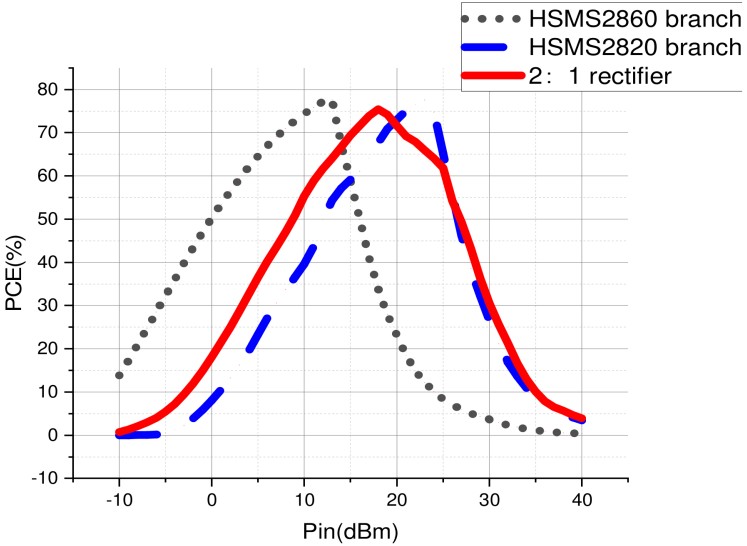

**Figure 12.** Comparison of the dynamic power range.

With the purpose of further demonstrating the feasibility of this technology, a mathematical model was established to analyze the fundamental principles of this technology. The approximation method was adopted, whereby the efficiency–power curves of diode 1 and diode 2 were assumed to possess an equilateral triangle structure. The maximum rectification efficiency of diode 1 and diode 2, occurred at input power Pin1 = 3 mW and Pin2 = 6 mW, as plotted in Figure 13.

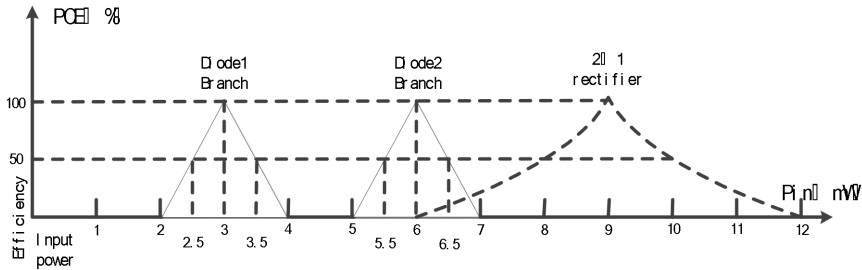

**Figure 13.** The mathematical model of the wide-dynamic dual RF-DC microwave rectifier circuit.

It can be observed from the figure that the dynamic power ranges with efficiencies higher than 50% at both diode 1 and diode 2 are 1 mW. To calculate the dynamic power range for rectification efficiencies higher than 50% in a 2:1 dual RF-DC rectifier circuit, the input power, Pin, of the 2:1 dual RF-DC microwave rectifier circuit was calculated as 3x (unit mW). Through the unequal 2:1 power divider, the input power values Pin1 and Pin2 obtained for the circuits of diode 1 and diode 2 should be x and 2x. Considering the function relationship of the equilateral triangle, this needs to be evaluated in different contexts.

1. When $P_{in} = 3x \in [0,6] \cup [12, +\infty]$, the input power of the circuit at diode 1 is $P_{in1} = x \in [0,2] \cup [4, +\infty]$, while the input power of the circuit at diode 2 is $P_{in2} = 2x \in [0,4] \cup [8, +\infty]$. Under these conditions, the rectification efficiency of the two circuits should be PCE1 = PCE2 = 0, which is excluded in this paper.

2. When $P_{in} = 3x \in [6, 7.5] \cup [10.5, 12]$, the input power of the circuit at diode 1 is $P_{in1} = x \in [2, 2.5] \cup [3.5, 4]$. Under these conditions, the rectification efficiency of this circuit should be PCE2 $\leq$ 50%. The input of the circuit at diode 2 is $P_{in2} = 2x \in [4, 5] \cup [7, 8]$, so the rectification efficiency of this circuit should be PCE2 = 0. Moreover, because Pin1 < Pin2, the total efficiency of the 2:1 dual RF-DC rectifier circuit is PCE < 50, which is excluded in this paper.

3. When $P_{in} = 3x \in [7.5, 9]$, the input power of the circuit at diode 1 is $P_{in1} = x \in [2.5, 3]$, and the input power of the circuit at diode 2 is $P_{in2} = x \in [5, 6]$; thus, the total efficiency of the 2:1 dual RF-DC rectifier circuit can be calculated on the basis of the following formula:

4. $PCE = \frac{x \cdot \sqrt{3}(x-2)/\sqrt{3} + 2x \cdot \sqrt{3}(2x-5)/\sqrt{3}}{3x} \times 100\% \geq 50\%$

5. $P_{in} = 3x \geq 8.1 \text{mW}$

6. When $P_{in} = 3x \in [9, 10.5]$, the input power of the circuit at diode 1 is $P_{in1} = x \in [3, 3.5]$ and the input power of the circuit at diode 2 is $P_{in2} = x \in [6, 7]$; the total efficiency of the 2:1 dual RF-DC rectifier circuit can be calculated on the basis of the formula below:

$$PCE = \frac{x \cdot \sqrt{3}(4-x)/\sqrt{3} + 2x \cdot \sqrt{3}(7-2x)/\sqrt{3}}{3x} \times 100\% \geq 50\%$$
$$P_{in} = 3x \leq 9.9 \text{mW}$$

It can be observed from the mathematical derivation described above that the dynamic power range of the 2:1 dual RF-DC rectifier for efficiency above 50% is $P_{in} = 3x \in [8.1, 9.9]$mW, reaching 1.8 mW. Compared to the single microwave rectification achieved by each circuit, this represents an increase of 0.8mw, which can be attributed to the 2:1 dual RF-DC microwave rectifier circuit overcoming the limitations of the individual diode rectification circuits with respect to high-power input response capacity. High power is inputted into the HSMS2820 Schottky diode, which has stronger high-power input response ability, while low power is inputted into the HSMS2860 Schottky diode, which possesses a good response to low power, thus fully taking advantage of the different diodes' rectification capacity for the applicable dynamic input ranges in order to further optimize overall circuit efficiency and maintain the stability of total circuit rectification efficiency over a larger dynamic range of input power. This is highly significant for MWPT system engineering applications.

## 5. Conclusions and Prospects

A dual RF-DC microwave rectifier circuit based on an unequal Wilkinson power divider was designed in this paper. This rectifier circuit was designed with two input power components in the ratio of 2:1, passing through HSMS2820 and HSMS2860 Schottky diodes with different power response characteristics. Assuming optimized impedance matching for each rectifier circuit, we compared the overall dynamic power range of the circuit with the dynamic power range of each individual circuit. Compared to the HSMS2820 circuit, the dynamic power ranges above efficiency values of 60% and 50% were increased by 4 dBm and 3 dBm, respectively. Compared to the HSMS2860 circuit, the dynamic power ranges above efficiency values of 60% and 50% were increased by 5 dBm and 2 dBm, respectively, indicating that, relative to traditional rectifier circuits, the 2:1 RF-DC microwave rectifier circuit is better able to increase the dynamic power range while enabling the MWPT system to maintain relatively stable RF-DC transfer efficiency in constantly changing environments. Finally, a typical mathematical model was constructed to demonstrate the rationality of this conclusion.

**Author Contributions:** Conceptualization, C.P.; Data curation, C.P. and C.C.; Formal analysis, C.P. and J.W.; Funding acquisition, Z.Y. and C.C.; Investigation, C.P. and Z.Y.; Methodology, C.P.; Project

administration, C.P. and C.C.; Resources, Z.Y. and Z.W.; Software, J.W. and Z.W.; Writing—review & editing, J.W. All authors have read and agreed to the published version of the manuscript.

**Funding:** This research received no external funding.

**Conflicts of Interest:** The authors declare no conflict of interest.

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
