# Peer review of "Design of a Wide-Dynamic RF-DC Rectifier Circuit Based on an Unequal Wilkinson Power Divider"

_electronics, doi:10.3390/electronics10222815_

Round 1
Reviewer 1 Report
- This paper is about micro-scale energy harvesting systems and maximum power efficiency, so popular publications need to be added as references, such as
[1] Chao Lu, Vijay Raghunathan, Kaushik Roy, "Efficient Design of Micro-Scale Energy Harvesting Systems", IEEE Journal on Emerging and Selected Topics in Circuits and Systems, vol., 1, no. 3, pp. 254-266, 2011.
[2] Vijay Raghunathan, Pai Chou, "Design and power management of energy harvesting embedded systems", Proceedings of the 2006 international symposium on Low power electronics and design, pp. 369-374, 2006.
[3] Jun Yi, Wing Hung Ki, Chi Ying Tsui, "Analysis and design strategy of UHF micro-power CMOS rectifiers for micro-sensor and RFID applications", IEEE Transactions on Circuits and Systems I: Regular Papers, vol. 54, no. 1, pp. 153-166, 2007.
[4] Chao Lu, Vijay Raghunathan, Kaushik Roy, "Micro-scale energy harvesting: A system design perspective", 15th Asia and South Pacific Design Automation Conference (ASP-DAC), pp. 89-94, 2010.
2. It is necessary to compare this work with other existing works in the literature, so that the benefits and novelty of this work is clearly highlighted. Therefore, I suggest the authors to add a subsection or a paragraph to discuss this aspect.
3. When the input signal strength changes, how to adapt your circuit and system to track the maximum output power points? Do you consider the following methods (even though these methods are not for RF-DC conversion, but their idea is applicable to RF-DC cases)?
[1] Chao Lu, Chi Ying Tsui, Wing Hung Ki, "Vibration Energy Scavenging System With Maximum Power Tracking for Micropower Applications", IEEE Transactions on Very Large Scale Integration (VLSI) Systems, vol. 19, no. 11, pp. 2109-2119, 2011.
[2] Dexin Li, Pai Chou, "Maximizing efficiency of solar-powered systems by load matching", Proceedings of the 2004 International Symposium on Low Power Electronics and Design , pp. 162-167, 2004.
[3] Chien Ying Chen, Pai Chou, "DuraCap: A supercapacitor-based, power-bootstrapping, maximum power point tracking energy-harvesting system", 2010 ACM/IEEE International Symposium on Low-Power Electronics and Design, pp. 313-318, 2010.
[4] Chao Lu, Sang Phill Park, Vijay Raghunathan, Kaushik Roy, "Low-overhead maximum power point tracking for micro-scale solar energy harvesting systems", 25th International conference on VLSI design, pp. 215-220, 2012.
Reviewer 2 Report
In this paper, a dual-channel RF-DC microwave rectifier circuit with a 2:1 power distribution ratio of Wilkinson power splitter is designed. The rectifier circuit works at 2.45ghz. After impedance matching and tuning, the structure can broaden the dynamic power range of the rectifier circuit while maintaining the maximum rectifier efficiency.Overall, this paper is interesting,
The main comments are:
1. The title of RF-DC rectifier circuit and microwave rectifier circuit have the same meaning
2. Some literatures are quite old, so it is suggested to retain the most classic literatures and add some latest literatures
3. It would be more convincing to suggest carrying out some follow-up experimental work
Reviewer 3 Report
In this paper a Microwave Rectifier based on Unequal Wilkinson Power Divider is designed. This circuit contains passive parts and lumped elements together, which is interesting work. It needs some essential modifications.
- The device is not fabricated, only simulations results for this circuits at microwave frequency range, which has 5 lumped elements, which are soldered is not acceptable. First fabricate the circuit and then Provide measurement results and compare with simulations.
- In the lecture review some similar rectifiers should be introduced and at the end provide a comparison table between your design and mentioned works.
- The novelty of design is not clear, simple unequal Wilkinson divider and simple filter are used. Provide the main contribution of the work Cleary.
- Figs 3 and 4 should be redrawn and improved the quality of figures. The dimension of all stubs should be indicated in Fig.3.
- The substrate specification should be included before first simulation results.
- Which software is used to simulation?
- The S-parameters (Fig4) of the filter is not clear. Is this band pass filer, which has narrow pass band near 1.22 GHz? This is not clear at all. The pass band frequency, FBW, insertion loss and return loss parameters of the filter should be clearly, reported in the text.
- The rectifier output curves like return loss and efficiency should be reported versus frequency as separate figures.
- Which is the orating frequency of the design work? There is a conflict in design. The divider works at 2.45 GHz and filter eliminate this frequency???? The filter eliminates 2.45 GHz with more than 40 dB attenuation.
Round 2
Reviewer 3 Report
The authors answer some questions in the revised version but some comments and questions are still remained without any corrections.
- The fabricated circuit is necessary for this design, if authors cannot implement the design, At least simulations in two environments of EM (momentum) and schematic (circuit) in ADS software and comparison of these two simulations results are required.
- The quality of figures and curves are not acceptable at all, specially ADS output curves, which should be modified or redrawn with drawing software like Microsoft Visio.
- Improved the literature for harmonics suppressing parts. Described harmonics control circuit (HCC) in [R1-R2] can be useful. This comment is optional if you think this is helpful add these two references.
[R1] Design of a high efficiency Class-F power amplifier with large signal and small signal measurements. Measurement, 149, p.106991. 2020.
[R2] A modified Class-F power amplifier with miniaturized harmonic control circuit. AEU-International Journal of Electronics and Communications, 97, pp.202-209. 2018.
